# Associations between Vertebral Localized Contrast Changes and Adjacent Annular Fissures in Patients with Low Back Pain: A Radiomics Approach

**DOI:** 10.3390/jcm12154891

**Published:** 2023-07-25

**Authors:** Christian Waldenberg, Helena Brisby, Hanna Hebelka, Kerstin Magdalena Lagerstrand

**Affiliations:** 1Department of Medical Radiation Sciences, Institute of Clinical Sciences, Sahlgrenska Academy, University of Gothenburg, 413 45 Gothenburg, Sweden; kerstin.lagerstrand@vgregion.se; 2Institute of Clinical Sciences, Sahlgrenska Academy, University of Gothenburg, 405 30 Gothenburg, Sweden; helena.brisby@vgregion.se (H.B.); hanna.hebelka@vgregion.se (H.H.); 3Department of Medical Physics and Biomedical Engineering, Sahlgrenska University Hospital, 413 45 Gothenburg, Sweden; 4Department of Orthopaedics, Sahlgrenska University Hospital, 413 45 Gothenburg, Sweden; 5Department of Radiology, Sahlgrenska University Hospital, 413 45 Gothenburg, Sweden

**Keywords:** intervertebral disc, vertebrae, annular fissure, low back pain, radiomics, texture analysis, artificial intelligence (AI), Modic changes

## Abstract

Low back pain (LBP) is multifactorial and associated with various spinal tissue changes, including intervertebral disc fissures, vertebral pathology, and damaged endplates. However, current radiological markers lack specificity and individualized diagnostic capability, and the interactions between the various markers are not fully clear. Radiomics, a data-driven analysis of radiological images, offers a promising approach to improve evaluation and deepen the understanding of spinal changes related to LBP. This study investigated possible associations between vertebral changes and annular fissures using radiomics. A dataset of 61 LBP patients who underwent conventional magnetic resonance imaging followed by discography was analyzed. Radiomics features were extracted from segmented vertebrae and carefully reduced to identify the most relevant features associated with annular fissures. The results revealed three important texture features that display concentrated high-intensity gray levels, extensive regions with elevated gray levels, and localized areas with reduced gray levels within the vertebrae. These features highlight patterns within vertebrae that conventional classification systems cannot reflect on distinguishing between vertebrae adjacent to an intervertebral disc with or without an annular fissure. As such, the present study reveals associations that contribute to the understanding of pathophysiology and may provide improved diagnostics of LBP.

## 1. Introduction

Low back pain (LBP) is one of the most widespread disorders and is ranked highest in causing disability and societal costs globally [1,2]. The pathophysiological background of LBP is multifactorial, and regarding spinal tissue changes, degenerated intervertebral discs (IVDs) [3,4], pathological changes in the vertebra [5], and damaged cartilaginous and bony endplates [6,7] have been reported to be key risk factors. Specifically, annular fissures reaching the outer annulus fibrosus (AF) have been reported to be an entry point for the ingrowth of vascularized tissue and nociceptive nerve endings and participate in pain signaling [8,9]. In addition, vertebral bodies and endplates are highly innervated by nociceptors [10,11], which are densified in areas with damaged endplates. It has been suggested that such damage is associated with increased IVD degeneration and fissuring [12].

Overall decreased or locally impaired IVD nutrient supply due to damaged endplates is a recognized contributor to IVD degeneration [13], leading to altered extracellular matrix composition and weakened tissue strength [14]. Consequently, the disc tissue becomes prone to structural damage, e.g., fissuring, further propagating the degenerative progress [15]. Inversely, degenerated IVDs also may influence adjacent vertebrae. Although the etiology is unclear, it has been suggested that endplate bone marrow lesions, often referred to as Modic changes (MCs), result from inflammatory factors diffusing from adjacent IVDs at sites with endplate damage [16]. Such pro-inflammatory factors are abundant within the nucleus pulposus (NP) in degenerated discs [17], which may further drive the development of bone marrow lesions. Such crosstalk, i.e., tissue changes in the IVD that may affect the vertebra and vice versa, is confirmed by several histological studies [12,18]. However, previous studies using clinical radiological markers, i.e., MC, show inconclusive results [19,20,21], which is why more qualified markers of association would be valuable.

With superior contrast visualization, MR imaging is the primary tool used to establish the status of the spine. Earlier studies have exploited the MR contrast to study associations between degenerative disc changes and vertebral changes. However, they rely on human observers and gross radiological markers, such as high-intensity zones (HIZ) [8], MCs, and Pfirrmann categorization [21,22]. These categorization schemes are ordinal, forcing the observers to choose from a predefined set of explanations. Data-driven analysis of radiological images has recently found its way into the LBP research as it can offer a more detailed evaluation of the image contrast without the dependence on the observer [23,24,25]. Especially, texture analysis that can convert conventional MR images into quantitative features, so-called radiomics, has shown promise [26]. One key advantage of radiomics is its ability to provide an understanding of the basis for a possible association, as the method can reflect tissue changes that can be translated into pathological patterns. Furthermore, the features can reflect patterns and abnormalities not identified by subjective visual interpretation alone and relate those findings to pathology. As such, radiomics holds promise in identifying potential associations due to crosstalk between vertebrae and disrupted IVDs and can reveal image patterns that support the association.

By studying a unique imaging material of a cohort of LBP patients examined by conventional magnetic resonance imaging (MRI) and discography during the same day, this study focuses on determining possible associations between vertebral bone marrow lesions viewed on MRI and adjacent annular fissures in a clinical setting using radiomics.

## 2. Materials and Methods

The existing dataset was consecutively and prospectively collected between April 2007 and March 2010 initially to investigate the impact of spinal loading and disc degeneration on pain provocation at discography [27,28,29]. All current IVDs have previously been phenotyped to investigate the association between annular fissuring and LBP [30] and the predictive ability of radiomics to identify annular fissures in conventional MR images [24]. The present study investigates the association between vertebrae and adjacent IVDs using radiomics. Inclusion criteria were chronic LBP > 6 months with failed conservative therapy. Low-quality images (low signal-to-noise ratio, motion artifact) were excluded from the study.

### 2.1. Diagnostic Procedures and Imaging Protocols

Within one day, the lumbar spine (L1-S1) of each patient was examined in the following consecutive order: 1.5T MRI using clinically conventional protocols, including sagittal and axial T1-weighted (T1W) and T2-weighted (T2W) imaging, followed by low-pressure discography (<50 psi), and lastly, computer tomography (CT) (Table 1).

### 2.2. IVD and Vertebrae Tissue Grading

The sagittal and axial CT discograms were used to grade the extension of the fissures, according to the Dallas Discogram Description (DDD) [31]. Since mainly fissures reaching the outer AF inflict LBP [8,32], IVDs with fissures extending to the outer 1/3 of the AF (DDD = 2–3) were separated from the rest (DDD = 0–1). All vertebrae were categorized by a medical student, supervised by the senior radiologist, according to the Modic characterization system [33] using the T1W and T2W images.

### 2.3. Image Segmentation

Using ITK-SNAP v3.8.0 [34], a medical physicist, guided by a senior radiologist, manually segmented 5–7 midsagittal slices of the vertebrae adjacent to IVDs diagnosed with discography. The vertebral regions of interest (ROIs) were divided in half, forming superior and inferior units. The segmentation was performed on the T2W images, after which the delineations were transferred to T1W to enable feature extraction from the T1W images. To ensure proper ROI placement on the T1W images, the T1W and T2W volumes were first registered through rigid registration using DICOM header information. Manual adjustment of the ROIs was performed when necessary. In previous studies, similar vertebrae segmentation has shown excellent intra-observer agreement within our research group (ICC = 0.9–1.0) [35,36].

### 2.4. Radiomics

#### 2.4.1. Image Preprocessing

Image processing was performed using MATLAB R2022b (Mathworks, Natick, MA, USA) and PyRadiomics v3.0.1 [37] on Python v3.7:Interpolation—To ensure rotationally invariant features, the MR images were interpolated to isotropic voxels of size 1 × 1 × 1 mm^3^ [38].Normalization—Each MR image volume was normalized to the mean volume signal intensity ± 3 standard deviations to increase the stability and reproducibility of calculated features [39,40]. Image voxels falling outside the boundaries were truncated to the lowest/highest value.∙ Intensity discretization—Discretization of the image intensities inside the ROI was performed to reduce the image noise level [41,42] and allow for feature calculation [43]. Even though the Image Biomarker Standardization Initiative (IBSI) recommends intensity discretization using a fixed bin number for MR images [44], studies have shown that a fixed bin width approach produces more reproducible features [45,46] when the number of bins is kept between 32 and 128 [47,48]. As such, with regard to the intensity range present inside the ROIs of the current images, intensity discretization was performed using an appropriate fixed bin width of three.

#### 2.4.2. Feature Calculation and Standardization

All statistical and texture-based features available in the PyRadiomics-package were calculated for each ROI on the T1W and T2W images. As the study focused on the association between vertebral tissue pattern and IVD fissures, morphological shape features were not calculated. To increase the interpretability of the features, no features of filtered MR images were calculated.

As standardization of the calculated features has been shown to increase the robustness and prognostic power [49,50], each feature was standardized by the z-score formula, which transforms each feature to have a global mean of 0.0 and variance of 1.0 [51].

#### 2.4.3. Feature Reduction

Radiomics features have been shown to be sensitive to image acquisition and reconstruction parameters [52], inter- and intra-observer variations during segmentation [53], and are often subjected to high correlations indicating data redundancy [52]. As such, the number of features to be included in the statistical and machine learning models was reduced to generate generalizable results [54]. The feature reduction was applied in a multi-step process:The features’ robustness to the variability of ROI segmentation was investigated by calculating future values using the initial vertebral segmentation and segmentation contracted by one pixel in all directions. The Intraclass Correlation Coefficient (ICC) using one-way random effects with absolute agreement, ICC(1, 1), was calculated using individual feature values as subjects and the two ROI perturbations as the raters. Features indicating poor reliability (ICC(1, 1) < 0.5) [55] were excluded from further analysis (see Appendix A). A similar methodology has been applied in recent studies [56].Similarly, features robustness to image acquisition and reconstruction was evaluated by interpolating the MR images into voxels of size 1 × 1 × 1 mm^3^ and 1.1 × 1.1 × 1.1 mm^3^ before feature calculation. ICC(1, 1) was calculated using individual feature values as subjects and the two voxel volume perturbations as the raters. Features with ICC < 0.5 were excluded from further analysis (Appendix A).Using the person correlation metric, features were pairwise tested for linear correlation. Pairs of features that displayed a very high linear correlation (R > 0.9) [57] were individually tested for correlation to the presence of fissure. The feature with the lowest correlation to the presence of a fissure was excluded from further analysis.The remaining features were further reduced through sequential backward feature selection algorithms to select the most meaningful features that reflect an association between feature and fissure. Three separate algorithms were used to predict an annular fissure in an adjacent IVD (Figure 1):
A fully connected neural network with one hidden layer with 100 nodes (Multilayer perceptron);A random forest ensemble of 100 trees built with bootstrap samples and balanced class weight;K-nearest neighbor classifier using five neighbors with uniform weights, i.e., all points in each neighborhood were weighted equally.

Due to imbalanced datasets (approximately four times more IVDs with fissures compared to IVDs without fissures), balanced accuracy was used as a scoring metric during the backward feature selection for all models. A 5-fold cross-validation was implemented to reduce the risk of overfitting. The top five most important features selected by each algorithm were selected, and the remainder were excluded from further analysis.

5.A binary logistic regression using backward elimination was applied to the top-performing features. The procedure establishes the importance of each feature to model fit, which reflects the association between features and fissures. Features that did not significantly contribute to the model fit (*p* > 0.05) were eliminated from further analysis.

**Figure 1 jcm-12-04891-f001:**
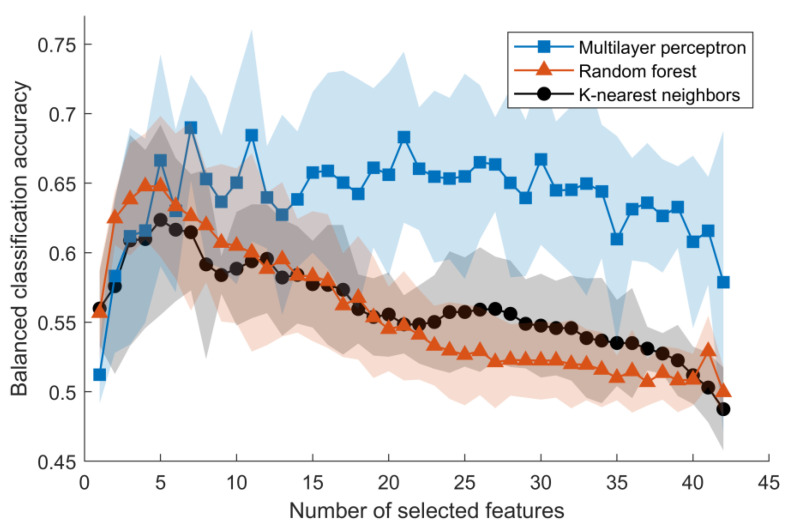
Balanced classification accuracy of identifying an annular fissure based on features calculated in an adjacent vertebra. The figure displays the mean accuracy ± one standard deviation (transparent colored area) calculated from 5-fold cross-validation. In two out of three models, the top five features were sufficient to achieve maximum classification accuracy.

### 2.5. IVD Fissure Association to Radiomic Features and Fissure Classification

The final model was based on a binary logistic regression fitted on the selected features that contributed significantly to model fit. To interpret associations between the vertebrae and adjacent IVD fissure, the beta value, *p*-value, and odds ratio was calculated for each feature.

### 2.6. Comparison between Radiomics and Radiological Markers of Vertebral Change

Previous studies have used clinical radiological markers i.e., MC, to study associations between vertebral changes and annular fissures but with inconclusive results [19,20,21]. For that purpose, a sub-study was included to put MC in perspective to radiomics.

To enable this comparison, classification results for associations between vertebrae and disc fissures were calculated both using the selected vertebrae features as well as MC to predict fissures, respectively.

### 2.7. Statistical Analysis

Statistical calculations were performed with MATLAB R2022b and IBM SPSS Statistics v29.0.0.0. The available data in the original studies determined the current patient sample size. The predictive ability of the logistic regression model using the selected features was evaluated in terms of sensitivity, specificity, and accuracy. Further, a receiver operating characteristic analysis was performed. The mid-*p*-value McNemar test at a 5% significance level was used to compare associations between IVD fissures and the selected features as well as MCs. That is, it was tested to identify whether the predictive accuracy of the logistic regression using the selected features was more accurate than using MC to identify an outer annular fissure. The intra- and inter-observer reliability/agreement for the grading of the DDD and segmentation were not evaluated here but in previous studies using current and other datasets [27,35,36].

## 3. Results

After excluding five patients due insufficient image quality, sixty-one patients (age 24–63 years, mean age 45 years ± 9 (standard deviation), 32 female) were included to study the association between the vertebral tissue patterns and adjacent annular fissures (Table 2).

### 3.1. Feature Dimensionality Reduction

One hundred and eighty-six first-order features and texture features were calculated from the T1W and T2W images of vertebrae. Five features were not robust to segmentation variability or displayed poor reliability (ICC(1, 1) < 0.5) [55] and were excluded from further analysis, and 139 features displayed very high linear correlation (R > 0.9) [57] and were excluded, leaving 42 features (see Appendix A). The remaining features were processed through the sequential backward feature selection algorithms to select the most important features. The top five features selected for each model were exclusively texture features (Table 3).

The binary logistic regression using backward elimination eliminated two out of five features that did not significantly contribute to the model fit. The top three highly important features that probe variations in vertebral image contrast were the following:gldm_LargeDependenceHighGrayLevelEmphasis_t1w (LDHGLE);glszm_LargeAreaHighGrayLevelEmphasis_t1w (LAHGLE);glszm_SmallAreaLowGrayLevelEmphasis_t1w (SALGLE).

Specifically, LDHGLE emphasizes high gray levels with strong spatial dependence, LAHGLE highlights large areas with high gray levels, and SALGLE accentuates small areas with low gray levels.

### 3.2. Association between Vertebra and Adjacent IVDs

The negative feature beta values (B) in the logistic regression model indicate that an increase in all or any feature values decreases the probability of an adjacent IVD having an outer fissure (Table 4). All three selected values significantly contributed to model fit (*p* ≤ 0.002).

Each of the selected features differed significantly between groups, i.e., the data in each group came from populations with unequal means (*p* ≤ 0.043). The largest overlap was found for the feature SALGLE. The feature maps of the selected features emphasized homogeneous regions in the vertebrae that were either large with high intensity (LDHGLE and LAHGLE) or small with low intensity (SALGLE) (Figure 2).

### 3.3. Fissure Classification

With the prediction threshold score set to 0.54 to maximize accuracy, the logistic regression correctly classified 291 out of 354 IVDs (each IVD was classified two times as it is adjacent to both superior and inferior vertebral units). The model reached a sensitivity of 96.5%, a specificity of 27.8%, and an accuracy of 82.5% (Figure 3). The ROC analysis displayed an area under curve (AUC) of 0.760, indicating acceptable discriminating ability [58].

Using MC to predict outer annular fissures yielded a sensitivity of 14.5%, a specificity of 100%, and an accuracy of 31.9% (Figure 4). The logistic binary regression model was significantly more accurate compared to using MC as a marker for an annular fissure (*p* < 0.001).

## 4. Discussion

This radiomics study suggests evidence of crosstalk between IVD fissuring and adjacent vertebrae, supported by the strong association between the discography-identified annular fissures and the MRI contrast features. The study may improve the basic understanding of what specific MRI contrast patterns in the vertebrae are associated with annular fissures. The classification results, achieved through the logistic regression, further confirmed the close relationship and the presence of crosstalk.

In the study, three textural features were identified that depicted the MR signal variations in the vertebrae which were associated with crosstalk between IVD fissures and adjacent vertebrae. It is important to address that the selected features do not directly reflect different types of tissues, but the behavior of the tissue is translated into image contrast. The features LDHGLE and LAHGLE emphasize large homogeneous bright areas on the T1W image. For these features, it was found that an IVD with a fissure was likely to be adjacent to a vertebra with low LDHGLE or LAHGLE feature values, reflecting an inhomogeneous vertebra with few or small bright areas (see example in Figure 2, row 2–4). In the T1W images, fat is displayed with a high signal where fattening of the bone marrow gives a distinctly higher signal in relation to the surrounding normal bone marrow. Hence, the absence of large bright areas (i.e., normal fatty bone marrow) might reflect that such vertebrae are affected mainly by edema or sclerotic components. Inversely, high feature values reflected a homogeneous vertebra with a little signal variation. IVDs without fissures were likely to be adjacent to such vertebrae (see example in Figure 2, top row). The third feature that the model selected to be of importance, i.e., SALGLE, has the property to emphasize small dark regions in MR images. In this study, the feature emphasized small dark regions on the T1W image that might correspond to sclerosis (Figure 2, column 3). Further, these findings, displaying a negative beta-value in logistic regression model, suggest that the absence of such pathology is more common in vertebras that are adjacent to IVDs with fissures. Although the causality is unclear, summarizing these three features, it is apparent that the presence of an adjacent inhomogeneous vertebrae seem to be an important factor for having a disrupted annular fissure.

Another interesting finding was that the T1W image was superior to the T2W image in reflecting a contrast behavior of importance for the crosstalk association. In fact, features extracted from the T2W image contributed with no additional information when the features from the T1W image were included in the analysis. The final most important features that the model selected only probed the T1W image, where both edema and sclerosis appeared dark. This might reflect a pathological process and might imply that the differentiation of edema from sclerosis is not of high importance for the crosstalk association. Dudli S. et al. concluded that inflammatory factors diffusing from degenerated IVDs might induce bone marrow lesions with active inflammation and edema [16], appearing dark on T1W images. This and the fact that Jensen T.S et al. have shown that these changes can directly convert to low signal sclerotic tissue [59] supports our findings.

There is no pathophysiological explanation or histological evidence indicating that MC should be directly linked to the presence of annular fissures, even if they often co-exist in a spinal motion segment. The actual pathology is more complex than the MC system categorizes, for example, the vertebra might be affected by bone marrow lesions with a mix of states, both an active inflammation (categorized into MC type 1) and fatty replacement of the marrow (categorized into MC type 2). As a result, these radiological markers are relatively insensitive and unspecific [24], which might explain the limited association between IVD disruption and MCs in current and other studies [19,20]. Furthermore, this suggests the need for more specific markers.

Medical data analysis has advanced significantly, with a recent focus on AI-driven techniques. Classical machine learning algorithms used here are well established through extensive study and development. What is more important is that they often offer the advantage of providing interpretable information, allowing clinicians and researchers to better understand underlying patterns that contribute to a certain outcome. Further, the present radiomics features used as input to the classical machine learning algorithms are based on well-defined mathematical equations, providing them with a solid theoretical foundation that can be used to explain image characteristics in a logical and derivable manner. The features provide valuable image information from routine clinical scans in a quantitative form, which potentially reflect underlying molecular changes of diseases at a cellular level [30,43]. By using such data-driven features, the analysis will not be limited by the human ability to detect complex and subtle image changes, and from these observations, draw conclusions. Thus, the combination of machine learning models with radiomics, will not eliminate the need for human interpretation but provide more comprehensive insights into medical imaging data and may have the potential to contribute to the development of trustworthy precision medicine.

The clinical value of diagnosing annulus fissures and tissue changes of the vertebrae for individual LBP patients is still unclear. However, increased knowledge of the relationship between different tissue injuries and the possibility of detecting small tissue changes, e.g., annular fissures, without invasive methods, may assist in our understanding of the causes of the pain in chronic LBP patients and should be evaluated further in normative populations to assess possible differences in prominent radiomic features.

Advanced image analysis methods are powerful tools to explore such different patterns of pathological changes that, in the next step, may be linked to specific symptoms, aiming to sharpen the diagnostics on an individual plan. However, the methods used in this study include both strengths and limitations. The design builds upon using a reliable but invasive diagnostic procedure (discography) to depict the annular fissures, which is a strength. However, due to its concomitant side effects and questionable reliability/reproducibility in detecting provoked pain [28,60], such diagnostics are seldom used in clinical practice nowadays, limiting the possibility of repeating the study. Furthermore, the proposed data-driven method holds great promise as it may be adopted into other research topics to explore associations with the possibility of increasing the understanding of causality.

## 5. Conclusions

The present study suggests that radiomics can objectively describe heterogeneity patterns in the vertebral tissue associated with adjacent disrupted annular fissures. The characteristic image patterns described by the radiomics features may improve the understanding of this crosstalk association and the quantitative nature of radiomic features enable objective longitudinal studies of LBP patients.

## Figures and Tables

**Figure 2 jcm-12-04891-f002:**
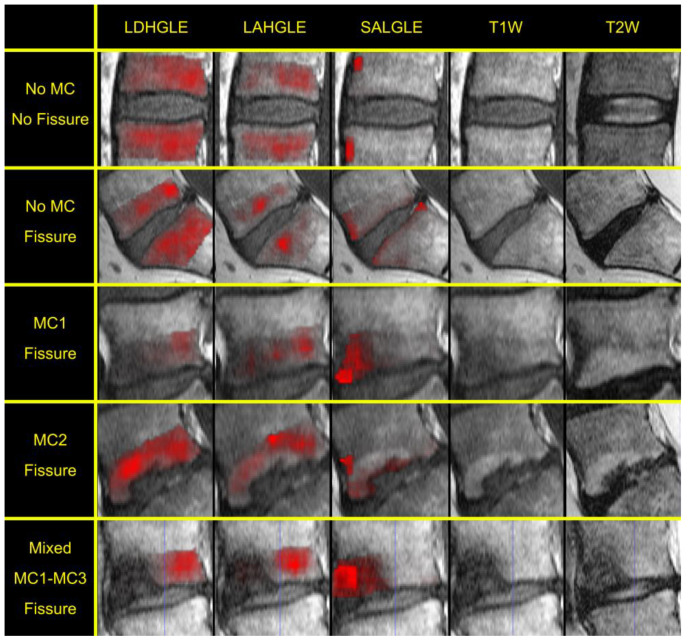
Feature map examples overlaid onto T1W images of the vertebra. In the examples, the logistic regression correctly classified the adjacent IVD. The selected feature values were calculated from T1W images; as such, the feature maps are overlaid on the T1W image. For reference, T1W and T2W image, without feature map overlay, is included. Feature maps are generally more homogeneous over larger areas when calculated from vertebra with no adjacent IVD fissure (top row) and have a more regional focus and irregular texture when calculated from vertebra with an adjacent IVD with an annular fissure.

**Figure 3 jcm-12-04891-f003:**
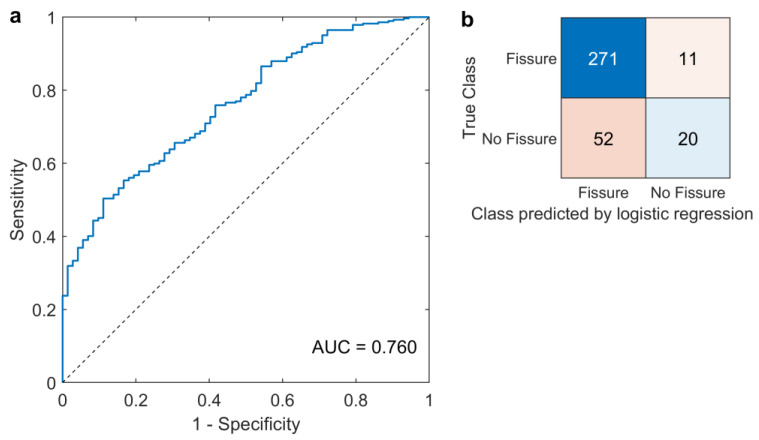
(**a**) Receiver operating characteristic (ROC) visualizing model sensitivity and specificity at different prediction cut-off scores. Area under curve (AUC) value of 0.760 was reached. (**b**) Confusion matrix displaying the classifying performance of the logistic regression using the selected features as a marker to identify an outer annular fissure in an adjacent intervertebral disc. The number of true positives (top left), true negatives (bottom right), false positives (bottom left), and false negatives (top right) is presented at a 0.537 cut-off score.

**Figure 4 jcm-12-04891-f004:**
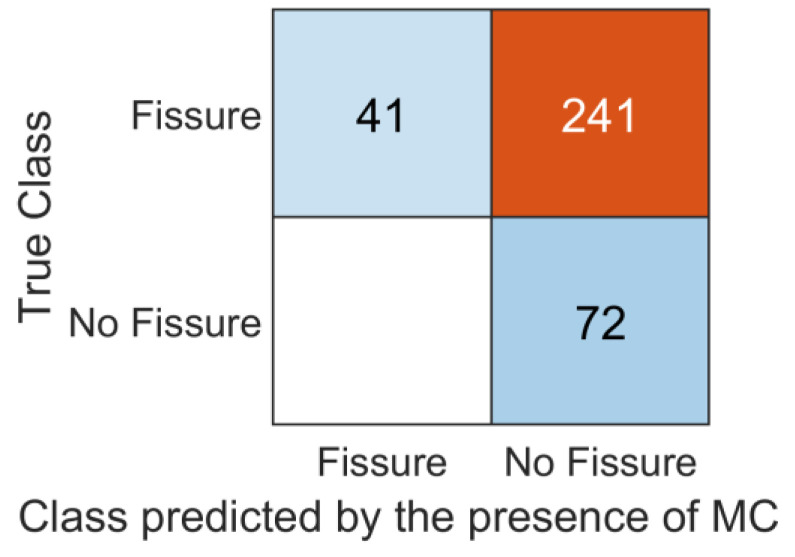
Confusion matrix displaying the classifying performance of MC as a marker to identify an outer annular fissure in an adjacent intervertebral disc. The number of true positives (top left), true negatives (bottom right), false positives (bottom left), and false negatives (top right) is presented.

**Table 1 jcm-12-04891-t001:** Scan and reconstruction parameters for MRI and CT protocols.

Parameter	T1W MRI (TSE) ^a^	T1W MRI (SE) ^a^	T2W MRI (TSE) ^a^	T2W MRI (TSE) ^a^	CT ^b^
Imaging plane	Sagittal	Axial	Sagittal	Axial	Sagittal, Axial
Repetition time (ms)	448	500	4862	5000	
Echo time (ms)	11	15	97	119	
Echo train length	9	1	21	25	
Slice thickness (mm)	4.0	4.0	4.0	4.0	0.75 (reconstructed)
Slice gap (mm)	0.4	0.4	0.4	0.4	
Number of averages	4	2	2	4	
Pixel bandwidth (Hz)	200	100	190	190	
Flip angle (degree)	149	90	150	150	
Acquisition matrix	512 × 256	256 × 135	512 × 256	256 × 126	
Reconstruction matrix	512 × 512	384 × 512	512 × 512	360 × 512	512 × 512
Field of view (mm^2^)	300 × 300	135 × 180	300 × 300	127 × 180	162 × 162
Convolution kernel					B45s

^a^ MRI system: 1.5T Siemens Magnetom Symphony Maestro Class, Erlangen, Germany. ^b^ CT system: Siemens Somatom Sensation 16-slice, Erlangen, Germany. T1W = T1-weighted; T2W = T2-weighted; MRI = magnetic resonance imaging; CT = computer tomography; SE = spin echo; TSE = turbo spin echo.

**Table 2 jcm-12-04891-t002:** Demographic and radiographic characteristics of included patients.

Patient and IVD Characteristics		
Age (years)		45 ± 9 ^a^
No. of patients		61
No. of female		32 (52%)
No. of Modic Changes		41 (12%) ^b^
No. of IVDs		177
IVD segment	L1–L2	2 (1%)
	L2–L3	21 (12%)
	L3–L4	57 (32%)
	L4–L5	57 (32%)
	L5–S1	40 (22%)
Dallas Discogram Description	Grade 0–1	36 (20%)
	Grade 2–3	141 (80%)

Note. Except where indicated, data are numbers of IVDs. ^a^ Numbers are presented as the mean value ± one standard deviation. ^b^ The number in parentheses refer to percentages of the total number of vertebral units (2 times the number of IVDs). IVD = intervertebral disc.

**Table 3 jcm-12-04891-t003:** Top five features selected for each model using sequential backward feature selection.

Feature Name	Random Forest	K-Nearest Neighbors	Multilayer Perceptron
gldm_LargeDependenceHighGrayLevelEmphasis [t1w]	●	●	●
glszm_LargeAreaHighGrayLevelEmphasis [t1w]	●	●	
glszm_SizeZoneNonUniformity [t1w]	●		●
glszm_SmallAreaLowGrayLevelEmphasis [t1w]		●	
glszm_LargeAreaHighGrayLevelEmphasis [t2w]		●	
glszm_ZonePercentage [t2w]	●		●
glszm_ZoneVariance [t2w]	●		●
ngtdm_Coarseness [t2w]			●
ngtdm_Strength [t2w]		●	

t1w = T1-weighted; t2w = T2-weighted.

**Table 4 jcm-12-04891-t004:** Logistic regression summary.

Selected Features	B	Significance	Exp(B), (Odds Ratio)
gldm_LargeDependenceHighGrayLevelEmphasis_t1w	−0.98	<0.001	0.38 (0.26 0.56) ^a^
glszm_LargeAreaHighGrayLevelEmphasis_t1w	−0.66	<0.001	0.52 (0.38 0.69) ^a^
glszm_SmallAreaLowGrayLevelEmphasis_t1w	−0.62	0.002	0.54 (0.36 0.80) ^a^
Constant (intercept)	1.63	<0.001	5.11

^a^ 95% confidence interval. B = beta value; t1w = T1-weighted.

## Data Availability

The data presented in this study are available on request from the corresponding author. The data are not publicly available due to the confidentiality of the human participants.

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
