# Peer review of "Associations between Vertebral Localized Contrast Changes and Adjacent Annular Fissures in Patients with Low Back Pain: A Radiomics Approach"

_jcm, 2023, doi:10.3390/jcm12154891_

Round 1
Reviewer 1 Report
The present study is very actual and suggests that radiomics can objectively describe heterogeneity patterns in the vertebral tissue associated with adjacent disrupted annular fissures.
The title of the article matches the content. The purpose and objectives of the work are fully realized.
Figures and tables are very clear and informative.
Reference is formed according to the rules.
Discussion and conclusions follow logically from the results of the study and are fully consistent with the purpose of the study.
The main findings as related to the overall purpose of the study are discussed and explained in detail.
Conclusions is directly related to the data that was collected and analyzed.
Author Response
No adjustments to the manuscript were suggested.
Reviewer 2 Report
This is a very well-written piece of work that re-analyses an existing data base to explore associations between contrast changes and annular fissures in LBP patients. This is a very credible contribution to ongoing efforts to find mechanical biomarkers for LBP, and highlights the value of modern techniques and imaging in this process. Whilst of course the authors can only comment on association and not causation, the paper highlights some potentially interesting areas for further exploration and study.
There are just some minor amendments that I think would add value to the paper below.
Introduction
In the following line….’This additionally supports the presence of crosstalk and causality between vertebral changes and IVD disruption’. Can you comment on causality? There does not seem evidence to do so. Please also explain in more detail what you mean by cross-talk as this may not be clear to readers.
In relation to the following…. ‘Data-driven analysis of radiological images has recently found its way into the LBP research as it can offer a more detailed evaluation of the image contrast without the dependence on the observer’. It would be helpful here to provide references to support the statement.
In relation to ‘determining possible associations between vertebral changes and adjacent annular fissures in a clinical setting using radiomics’. This aim could be clearer, please re-state what ‘vertebral change specifically you are referring to’, I know there were a large number of variables to begin with at least, but perhaps you could provide some further clarity here.
Materials and Methods
With regards to ‘Selected Features’ e.g. gldm_LargeDependenceHighGrayLevelEmphasis_t1w. It may be useful to orientate the reader to explain in more detail what these features are.
Conclusion
It may be best to avoid comment on causation, and stick with association here. i.e. ‘it could be used to objectively evaluate the specific vertebral tissue changes caused by a near IVD pathology’. It is not clear from the paper that anything other than association has been shown.
Other comments
Although I realise this was a re-analysis of an existing data set, these features will also be prominent in many non-LBP individuals. It would have been interesting of course to have a normative population to compare the results with, this could perhaps be commented on.
There is perhaps an argument to say a ‘splatter-gun approach’ should be discouraged when looking for associations, however in this instance it is clear you have made great efforts to ensure you are focusing on a rational approach to the collective use of data.
This type of approach should be applauded, and will hopefully lead to advances in this area going forwards.
